# Macrostructural and Microstructural White Matter Alterations Are Associated with Apathy across the Clinical Alzheimer’s Disease Spectrum

**DOI:** 10.3390/brainsci12101383

**Published:** 2022-10-13

**Authors:** Riccardo Manca, Sarah A. Jones, Annalena Venneri

**Affiliations:** 1Department of Life Sciences, Brunel University London, Uxbridge UB8 3BH, UK; 2Rotherham Doncaster and South Humber NHS Foundation Trust, Rotherham DN4 8QN, UK; 3Department of Medicine and Surgery, University of Parma, 43126 Parma, Italy

**Keywords:** Alzheimer’s disease, apathy, white matter

## Abstract

Apathy is the commonest neuropsychiatric symptom in Alzheimer’s disease (AD). Previous findings suggest that apathy is caused by a communication breakdown between functional neural networks involved in motivational–affective processing. This study investigated the relationship between white matter (WM) damage and apathy in AD. Sixty-one patients with apathy (AP-PT) and 61 without apathy (NA-PT) were identified from the Alzheimer’s Disease Neuroimaging Initiative (ADNI) database and matched for cognitive status, age and education. Sixty-one cognitively unimpaired (CU) participants were also included as controls. Data on cognitive performance, cerebrospinal fluid biomarkers, brain/WM hyperintensity volumes and diffusion tensor imaging indices were compared across groups. No neurocognitive differences were found between patient groups, but the AP-PT group had more severe neuropsychiatric symptoms. Compared with CU participants, only apathetic patients had deficits on the Clock Drawing Test. AP-PT had increased WM damage, both macrostructurally, i.e., larger WM hyperintensity volume, and microstructurally, i.e., increased radial/axial diffusivity and reduced fractional anisotropy in the fornix, cingulum, anterior thalamic radiations and superior longitudinal and uncinate fasciculi. AP-PT showed signs of extensive WM damage, especially in associative tracts in the frontal lobes, fornix and cingulum. Disruption in structural connectivity might affect crucial functional inter-network communication, resulting in motivational deficits and worse cognitive decline.

## 1. Introduction

Apathy is commonly defined as a deficit in self-initiated goal-directed behaviors. However, this clinical label is also used to describe general loss of motivation/interest in social and cognitive activities, as well as blunted affect [1]. This neuropsychiatric symptom is the commonest in Alzheimer’s disease (AD), affecting about 50% of patients [2]. While the prevalence estimates of this symptom are highly variable (between 3% and 50%) in mild cognitive impairment (MCI) [3], apathy risk has been shown to increase as cognitive decline progresses [4] and to be associated with AD dementia severity more strongly than mood disorders [5]. The multiplicity of different definitions, operationalizations and assessment approaches for apathy have caused difficulties with its detection in patients with AD (and other neurodegenerative diseases). For these reasons, Miller et al. [6] sought expert consensus and proposed new diagnostic criteria for apathy across neurocognitive diseases. In brief, a patient who meets criteria for a diagnosis of a neurocognitive disorder, must present with significant and protracted (at least 4 weeks) behavioral alterations. They must display at least one of the following: (1) diminished initiative, (2) diminished interest or (3) diminished emotional expression/responsiveness that cannot be exclusively explained by other diseases and that must be causally linked to significant functional impairment.

Setting more comprehensive and internationally recognized criteria represents a first step to meeting clinical and research demands regarding diagnosis, effective treatment design and the deeper elucidation of this symptom across neurocognitive disorders of various etiologies. In fact, apathy is recognized as a marker of poor prognosis in people with AD, such as greater functional impairment, higher odds of institutionalization and increased mortality rates [7,8,9]. Moreover, apathy is also associated with faster and more severe cognitive decline in non-demented older adults [10], and it is said to lead to a sevenfold increase in the likelihood of progression from amnestic MCI to AD dementia [11]. Indeed, multiple studies have consistently highlighted greater executive function deficits in apathetic patients with AD on tests of set-shifting, attention and verbal fluency [12,13,14], as well as more severe closing-in errors in copying tasks, a phenomenon characterized by the tendency to draw near to or on top of the drawing model often observed in patients with dementia due to AD [15].

Recent investigations into AD biomarkers have shown that apathy severity is associated with decreased cerebrospinal fluid (CSF) levels of amyloid beta (Aβ) but not with concentrations of phosphorylated tau (p-tau) [16,17]. However, both high p-tau and low Aβ CSF levels have been found to be significant predictors of increased probability of apathy over time along the clinical AD continuum [18]. Consistently, positron emission tomography (PET) studies have reported that AD-related apathy is associated with increased Aβ accumulation in the orbitofrontal cortex (OFC) bilaterally and in the left superior frontal cortex [19] and increased p-tau accumulation in the left superior parietal cortex [20] and in the right anterior cingulate (ACC) and dorsolateral prefrontal cortices (PFC) [21]. Higher global brain levels of Aβ detected with PET have been found associated with increased likelihood of developing apathy over time [22], as well as with greater apathy severity [23] also in cognitively unimpaired older adults. This body of evidence appears to suggest that this neuropsychiatric symptom may be an early clinical marker of AD pathological changes. Indeed, a handful of *post mortem* investigations have found that apathy in patients with AD is associated with higher neurofibrillary tangle counts primarily in the ACC, while the contribution of pathological changes in other frontal and parietal regions remains unclear [24,25,26].

An influential pathophysiological model has hypothesized that dysfunction in the ventromedial PFC may cause impaired action–outcome assessment that would, therefore, represent the primary mechanism underlying AD-related apathy [27]. In detail, it has been suggested that disruption in the communication between the ACC, OFC and basolateral amygdala affects the transmission of a decision value signal to the nucleus accumbens. This defective transmission results in altered dopamine signaling to fronto-striatal circuits responsible for the execution of the most appropriate response. This model is supported by the findings of several neuroimaging investigations that have found an excess of hypometabolism [28] and structural alterations, i.e., greater gray matter (GM) volume loss, primarily in the ACC of patients with AD and apathy [29], as also highlighted by a meta-analysis [30]. However, different studies have reported apathy severity to be also associated with lower metabolism in the posterior cingulate cortex [31] and with GM atrophy in the OFC and in the left insula [32]. Moreover, one recent study that focused on cognitive apathy has revealed an association with atrophy in the right frontal pole and OFC, thalamus and putamen [33].

Considering that a functional disconnection across multiple networks, primarily involving frontal cortices, is the central tenet of the model proposed by Guimaraes et al. [27], more investigations have recently focused on assessing the relationship between the integrity of functional and structural connectivity and apathy in AD. In line with this model, resting-state functional alterations have been found primarily in the salience and fronto-parietal networks that have been reported to be less functionally segregated in patients with AD and apathy compared with non-apathetic patients [34]. In particular, the insular cortices appear to be less functionally connected and the dorsolateral PFC more connected with fronto-parietal regions in apathetic older adults both with [35] and without AD [36]. These findings seem to suggest that a combination of reduced salience processing and interoception, and increased fronto-parietal inhibition, consistent with the model by Guimaraes et al. [27], may explain the emergence of apathetic symptoms.

Widespread microstructural white matter (WM) alterations, in particular higher fractional anisotropy (FA) and lower mean diffusivity (MD), have also been reported to be associated with apathy presence and severity in people with amnestic MCI and AD dementia. These associations emerged more commonly in the cingulum [37,38], especially the anterior section close to the ACC [39,40,41], the superior longitudinal fasciculus and contiguous parietal WM areas [37,40,42,43], the corpus callosum [37,42,43,44], the anterior thalamic radiations and the uncinate fasciculus [37,38,40,42]. These findings appear to be far more heterogeneous than those from functional connectivity studies, but they are in line with the observation that patients with AD and evidence of WM damage (e.g., hyperintensities) are more likely to present with apathy [45,46]. In fact, apathy severity has been shown to be significantly associated with both greater global [47] and frontal WM hyperintensity (WMH) volume [48,49], but not when WM damage was assessed visually [50]. One voxel-lesion-symptom mapping study found that AD-related apathy was associated with damage to the anterior thalamic radiations [51].

Although a few studies seem to suggest that alterations in multiple WM tracts, especially in the frontal lobes, may lead to apathetic symptoms in AD, the paucity of investigations and the highly variable findings prevent any conclusions on what brain structural connections may be crucially involved in the genesis and/or persistence of this symptom. For these reasons, the primary aim of this study was to assess the relationship between apathy and WM damage by combining both macrostructural and microstructural analyses, in participants across the clinical spectrum of AD. This relationship was investigated by comparing WM volumes and WMH volumes, as measures of WM macrostructural alterations, and diffusion tensor imaging (DTI) indices, as measures of WM microstructural alterations, between patients with AD with and without apathy and cognitively unimpaired older adults. Considering the current literature, greater WM damage, especially microstructural alterations in frontal areas, was expected in apathetic compared with non-apathetic patients.

Since AD is a neurodegenerative disease primarily characterized by loss of GM tissue and gradual cognitive decline, a secondary aim of this study was to perform a set of complementary analyses to compare GM volume and cognitive performance across groups. Apathetic patients were expected to show greater GM loss, primarily in prefrontal and subcortical areas, potentially paralleled by more severe deficits in executive tasks.

## 2. Materials and Methods

### 2.1. Participant Sample

Participants were selected from the Alzheimer’s Disease Neuroimaging Initiative (ADNI) database (adni.loni.usc.edu, accessed on 1 October 2021). The ADNI was launched in 2003 as a public-private partnership, led by Principal Investigator Michael W. Weiner, MD. The primary goal of ADNI has been to test whether serial magnetic resonance imaging (MRI), positron emission tomography (PET), other biological markers, and clinical and neuropsychological assessment can be combined to measure the progression of mild cognitive impairment (MCI) and early Alzheimer’s disease (AD). For up-to-date information, see www.adni-info.org, accessed on 1 September 2022. The ADNI protocol was approved by the institutional review board of each site and all participants provided informed consent. This study performed secondary analyses on the ADNI dataset in compliance with the Declaration of Helsinki and ethical approval was granted by the Research Committee of Brunel University London (reference number 30422-TISS-Jul/2021-33453-2).

Initially, we screened all participants (*n* = 770) with an MRI assessment, including images to assess macrostructural and microstructural WM damage, i.e., T1-weighted images, fluid-attenuated inversion recovery (FLAIR) images and DTI scans. We screened all participants, independent of their diagnosis, who met the inclusion criteria based on the availability of data regarding: (1) behavioral alterations assessed by means of the Neuropsychiatric Inventory Questionnaire (NPI-Q) [52]; (2) severity of cognitive impairment assessed by means of the Clinical Dementia Rating (CDR) scale [53]; (3) global cognitive status assessed by means of the Mini Mental State Examination (MMSE); and (4) cognitive performance on a set of neuropsychological tests administered to the majority of ADNI participants, i.e., the Logical Memory Test, the Clock Drawing Test, the Auditory Verbal Learning Test, the Category Fluency Test (animals), the Trail Making Test and the Boston Naming Test. Participants who were classified as cognitively unimpaired (CU) were excluded if they presented with an NPI-Q score > 0. Details of the clinical, neuropsychiatric and cognitive assessments are available at http://adni.loni.usc.edu/methods, accessed on 1 September 2022.

After screening, a total of 224 participants met the inclusion criteria (i.e., had a comprehensive clinical assessment described above): 61 CU and 163 cognitively impaired (with a clinical diagnosis of either MCI or dementia due to AD). Cognitively impaired participants were classified as either apathetic (AP-PT, *n* = 77) or non-apathetic (NA-PT, *n* = 86), as recorded by the NPI-Q. Given the limited information that can be extracted from the NPI-Q data made available by ADNI, recently published diagnostic criteria for apathy in the context of neurocognitive disorders [6] could not be implemented.

Finally, 61 participants were selected from each patient group to match as closely as possible the CU group for age and education. The final sample used for this study included 183 participants divided into 3 groups: 2 patient groups, i.e., AP-PT (*n* = 61) and NA-PT (*n* = 61), and a control group of CU participants (*n* = 61). Data on cerebrospinal fluid (CSF) levels of biomarkers for AD, i.e., Aβ and p-tau as described in Toledo et al. (2013), sampled as close as possible to clinical assessment were available for only a subgroup of participants: 51/61 in the CU group, 53/61 in the NA-PT group and 58/61 in the AP-PT group. Participants were classified as positive to either biomarkers by using cut-offs calculated for the ADNI dataset: Aβ < 977 pg/mL and p-tau/Aβ ratio > 0.025 [54].

Information on medical history and on medications commonly prescribed to patients with AD and with potential effects on apathy severity, in particular acetylcholinesterase inhibitors and antidepressants [5,41,55], was also extracted for all participants included in this study.

### 2.2. MRI Pre-Processing

T1-weighted scans were re-oriented to the bi-commissural line and, subsequently, were pre-processed using a standard voxel-based morphometry (VBM) pipeline run with Matlab (Mathworks Inc., UK) and Statistical Parametric Mapping, version 12 (Wellcome Centre for Human Neuroimaging, London, UK): (1) scans were segmented to obtain three tissue maps, i.e., GM, WM and CSF; (2) GM and WM maps were normalized, i.e., warped and modulated, using a standard ICBM template in the MNI space; (3) finally, normalized scans were smoothed using an 8 mm Gaussian kernel. Global GM, WM and CSF volumes in ml for each participant were calculated in SPM12 following the procedure by Malone et al. [56]. Total intracranial volume (TIV) was calculated as the sum of the 3 global tissue volumes.

FLAIR images were also re-oriented to the bi-commissural line in SPM12. Subsequently, the Lesion Segmentation Toolbox (LST) v1.2.3 [57] was used to segment WMHs (i.e., lesions) by combining FLAIR and T1-weighted images [58]. The WMH segmentation threshold was set at *k* = 0.3 to quantify the total WM hyperintensity volume in ml in closest agreement with the gold standard lesion volume quantification procedure, i.e., manual segmentation [57]. Individual lesion probability maps were also generated and visually inspected to rule out misclassifications of lesion tissue. We preprocessed the WMH probability maps further by using a procedure suggested by the authors of the LST toolbox (https://www.applied-statistics.de/lst.html, accessed on 1 June 2022) and adapted from that of Mühlau et al. [59] to perform voxel-based analyses on WM lesions. First, the orientation of each individual map was checked and rectified in order to realign all images to the bi-commissural line. Second, T1-weighted images were lesion-filled using the LST toolbox. Third, lesion-filled T1-weighted images were segmented with SPM12, and deformation fields were saved. Fourth, each WMH probability map was normalized by applying the deformation fields obtained by segmenting the corresponding T1-weighted image. Finally, normalized maps were smoothed using a 6 mm Gaussian kernel.

DTI images underwent 3 preliminary pre-processing steps previously used [58] and implemented using the FMRIB Software Library v6.0.4 (FSL, http://www.fmrib.ox.ac.uk/fsl, accessed on 1 June 2022): (1) distortions caused by eddy currents and head motion were corrected for using the Diffusion Toolbox; (2) voxels of non-brain tissue were excluded by means of the Brain Extraction Tool and a 0.5 threshold was used to delineate the brain outline; (3) the diffusion tensor model was fitted at each voxel to obtain individual images for each DTI index of interest, i.e., FA, axial diffusivity (AxD) and mean diffusivity (MD). Radial diffusivity (RD) images were calculated as the average of L2 and L3 images automatically calculated by the diffusion tensor fitting.

The standard tract-based spatial statistics (TBSS) pipeline (http://fsl.fmrib.ox.ac.uk/fsl/fslwiki/TBSS/UserGuide, accessed on 1 June 2022) was used to complete the pre-processing of all the DTI index images. First, FA images were eroded to discard any potential outliers remaining after the diffusion tensor fitting. Second, FA images were non-linearly aligned to a standard template (FMRIB58_FA) and registered to the MNI152 standard space. Finally, a threshold of 0.2 was applied to the resulting 4D FA image to exclude GM and CSF voxels. AxD, MD and RD images, instead, were pre-processed using the “tbss_non_FA” script.

### 2.3. Statistical Analyses

Demographic, clinical and neural characteristics of the participant groups were compared using either ANOVA, for normally distributed variables, or the Kruskal–Wallis test, for non-normally distributed variables, and Bonferroni correction and Dwass–Steel–Critchlow–Flinger test were used for pairwise *post hoc* comparisons, respectively. Differences in rates of participants who were positive for AD biomarkers and in sex distributions were analyzed using the chi-square test. Analyses were carried out using SPSS version 26 (IBM, Chicago, IL, USA).

To address the primary aim of the study, three sets of voxel-based analyses were carried out using an ANCOVA model and three pairwise *post hoc* comparison models on WM maps, on WMH probability maps and on the four DTI indices. Moreover, three one-sample *t*-tests were performed on WMH maps in each participant group individually, to highlight areas of greater probability of macrostructural WM damage. All models included sex, TIV and an NPI-Q difference score (NPI-Q total score—apathy score) as covariates to control for potential differences across groups. WM and WMH VBM analyses were performed with SPM12 (cluster-level FWE-corrected *p* < 0.05), while DTI analyses were performed using the FSL tool “randomize” with 5000 permutations per model. Significant results were reported using threshold-free cluster enhanced (TFCE) images [60]. To extract peak and cluster data in MNI152 standard space, raw output images were masked with significant (*p* < 0.05) voxels from TFCE images.

To address the second aim, analogously to the analyses performed for clinical and demographic variables, we compared cognitive test scores across participant groups using either ANOVA or the Kruskal–Wallis test. Additional ANCOVA and pairwise *post hoc* VBM models were carried out on GM maps in SPM12 with the same covariates used in the WM analyses.

## 3. Results

### 3.1. Demographic and Clinical Variables

Patient groups had similar levels of global cognitive impairment, as detected by both CDR and MMSE total scores, and equivalent rates of positivity to both CSF biomarkers (i.e., Aβ and p-tau) (Table 1). Patients in the AP-PT group had mainly mild apathetic symptoms (*n* = 48) and only a minority had either moderate (*n* = 11) or severe apathy (*n* = 2). The AP-PT group included more men and had significantly higher NPI-Q scores (both total and after subtracting the apathy severity score) than the other two groups. Indeed, apathetic patients were also more likely to present with other neuropsychiatric symptoms (i.e., agitation, depression, anxiety, disinhibition, irritability, aberrant motor behaviors and appetite problems) than non-apathetic patients (Appendix A). Consistently, the AP-PT group was more likely to present with a history of mental health problems than the other participant groups (Appendix A), although no differences in rates of individual mental health diagnoses were observed between apathetic and non-apathetic patients (Appendix A). Indeed, apathetic patients presented with a significantly higher prevalence of antidepressant use, as well as of memantine, than the non-apathetic group (Appendix A).

Although the AP-PT group had a marginally larger TIV than the CU group (*p* = 0.049), comparable levels of global GM atrophy were highlighted in both patient groups, i.e., GMV/TIV values were significantly lower in both AP-PT (*W* = −6.54, *p* < 0.001) and NA-PT (*W* = −3.51, *p* = 0.035) than in the CU group. However, no differences in global WM volume were observed across groups, while global WM damage (i.e., WMHV/TIV values) was significantly higher, when compared with the CU group, only in the apathetic patient group (*W* = 4.07, *p* = 0.011; Figure 1).

### 3.2. Macrostructural and Microstructural WM Damage

No differences in voxel-based WM and WMH volumes were observed across groups in either the ANCOVA or the pairwise *post hoc* model. Indeed, all groups showed very similar patterns of WMH probability (Figure 2). However, the WMH maps generated by means of a VBM one-sample *t*-test for each group individually showed clusters of posterior WM lesions only in the patient groups. These appeared to be marginally larger in the AP-PT group.

The DTI analyses revealed significant differences in FA, AxD and RD between the AP-PT and CU groups (Table 2). Specifically, the AP-PT group presented with lower FA values primarily in the fornix, anterior thalamic radiations and uncinate fasciculus bilaterally, the left superior longitudinal fasciculus and right cingulum (temporal section), inferior fronto-occipital and longitudinal fasciculi (Figure 3A). The AP-PT group also had higher AxD values in the left superior longitudinal fasciculus (Figure 3B); and higher RD values in the fornix (Figure 3C).

Since more apathetic patients used memantine and antidepressants than non-apathetic patients, all AP-PT vs. NA-PT models were replicated including these variables as covariates. No additional findings emerged.

### 3.3. Cognitive Performance and Regional GM Volume

Both patient groups presented with deficits in most cognitive tests compared with the CU group. However, only apathetic patients showed significantly lower scores on the Clock Drawing Test (both copy and drawing) compared with the CU group (Table 3).

VBM ANCOVA analysis of GM maps showed significant differences. Both patient groups had similar patterns of GM atrophy to those of the CU group, primarily in bilateral medial temporal areas. The two patient groups showed a trend of differential GM atrophy of the cingulate gyrus: the anterior portion for apathetic patients, and the posterior portion for the non-apathetic group (Appendix A). In line with global GM volume analysis, no significant differences in voxel-based regional GM volumes emerged between AP-PT and NA-PT groups. The results were replicated after including the use of memantine and antidepressants as additional covariates in the AP-PT vs. NA-PT comparison.

## 4. Discussion

In this study, patients with MCI/dementia due to AD and apathy presented with a cognitive and neural profile similar to patients without apathy. However, those with apathy had a generally more compromised behavioral profile and were more likely to have a medical history of psychiatric conditions as well as to use antidepressants. Moreover, when comparing patient groups with a control group of matched CU older adults, only the AP-PT group presented with greater deficits on the Clock Drawing Test, larger global WM hyperintensity volume (i.e., macrostructural WM alterations) and alterations in DTI indices (i.e., WM microstructural damage).

DTI analyses showed that WM tracts connecting frontal and limbic areas were particularly affected in the AP-PT group. Both decreased FA and increased RD were found in the fornix of apathetic patients. This is in line with previous observations linking alterations in FA and MD in this fiber bundle with apathy in amnestic MCI [38], small vessel disease [61] and stroke [62]. The fornix primarily connects structures involved in episodic memory functions (i.e., the hippocampus and mamillary bodies), that are particularly affected in AD. This seems to suggest that the AP-PT group may show signs of more severe, although subtle, AD-related neuropathological alterations.

Although both patient groups were matched for severity of cognitive deficits across all tests, only the AP-PT group had worse cognitive performance on the Clock Drawing Test (both copy and drawing) than CU older adults. This test is considered to be a good screening tool for moderate/severe dementia [63], and our findings fit with the observation that apathy is associated with greater cognitive decline in this clinical population [11]. Our results are consistent with the notion that apathetic patients have more severe WM damage, as greater WM lesion volume [64] and hippocampal atrophy [65] have been found associated with increased Clock Drawing Test deficits in patients with AD.

Alterations were observed in both the anterior (higher voxel-based WM hyperintensity volume) and posterior-temporal (decreased FA) sections of the right cingulum in the AP-PT group. This WM tract is the one most consistently found associated with apathy in AD [37,38,39,40,41]. The cingulum is a complex associative WM tract that comprises multiple fibers connecting medial portions of various frontal, parietal and temporal cortices. Structural alterations in this tract have been reported in a wide variety of neurological and psychiatric conditions, as well as in association with apathetic symptoms [1,66]. The cingulum may indeed support multiple functions, related primarily to emotion/motivation processing in the anterior portion and to memory in the posterior section [66]. Therefore, the different cingulum microstructural alterations observed in apathetic patients may result in multiple aspects of motivational depletion, such as lack of interest or emotional reactivity. These were not assessed separately in this study.

Decreased FA was also observed in the apathetic group in the anterior thalamic radiations bilaterally and in the left SLF and forceps minor. All of these tracts connect different portions of the frontal lobes either to other frontal areas or to parietal and thalamic regions and support a range of cognitive control [67] and attentional functions [68]. Therefore, damage to these tracts may play a role in affecting decision-making and response selection processes that may contribute to a significant reduction in self-initiated goal-directed actions consistent with a state of apathy, as suggested by the model proposed by Guimaraes et al. [27]. This finding is consistent with previous observations reporting functional alterations in fronto-parietal networks [34,35,36] that have been interpreted as signs of enhanced cortical inhibition.

Moreover, the bilateral uncinate and inferior fronto-occipital fasciculi that connect frontal areas to the anterior temporal and occipital cortices [69], respectively, were also altered in the apathetic patients in this study, consistent with previous reports [37,38,42,61]. Considering the involvement of these WM tracts in emotional and attentional processes [69], these findings seem to support a conceptualization of apathy as a complex symptom underpinned by the potential combination of cognitive–emotional dysfunctions. Alterations observed in both the uncinate and inferior fronto-occipital fasciculi may also be consistent with a disruption to insular connections, since a previous study found a substantial overlap between these associative tracts and structural connections of three insular sub-regions [70]. The insula, in particular the left one, has been found to be more atrophic [32] and its functional connectivity with fronto-parietal circuits to be altered in apathetic patients with AD [35] and cognitively unimpaired older adults [36]. The insula is part of the salience network together with the ACC, an area considered to be particularly involved in motivation regulation [55]. Therefore, WM damage causing alterations in the communication between this network and fronto-parietal and thalamic areas, potentially mediated by the dopaminergic system [27], may increase the risk of apathy in this clinical population.

The findings of this study showed that the emergence of apathy in patients with AD appears to be primarily associated with an excess of WM damage, rather than of GM atrophy. Disruption to multiple WM tracts supporting the communication between medial (i.e., ACC) and dorsal PFC areas with the thalamus, parietal and temporal cortices may be playing a crucial role in driving deficits in goal-directed behaviors. Although no differences in either WM or GM were found between patient groups, only the apathetic patients presented with signs of extensive WM alterations when compared with CU older adults. The lack of significant differences in WM integrity between apathetic and non-apathetic patients is possibly due to the higher degree of variability observed in WM damage in the NA-PT (see Figure 1 that shows several outliers in the non-apathetic group) that may have masked potential between-group differences.

Patients with apathy were more likely to present with other neuropsychiatric symptoms, have a history of psychiatric problems and use antidepressants. In fact, depression has a high rate of comorbidity with apathy, although the two symptoms can present separately [71]. All analyses were covaried for an index of severity of neuropsychiatric symptoms other than apathy (NPI-Q total—apathy severity) and, thus, it could be reasonably argued that all significant results are genuinely associated with apathy. Moreover, the direct comparison between patient groups showed no significant differences in any neuroimaging outcome measure, even when all analyses were replicated including the use of memantine and antidepressants as covariates. Previous studies have also found that decreased FA in the cingulum [41] and hypometabolism in ACC/OFC and thalamus [28] were associated with apathy in AD independently of depression and medications with a possible impact on apathetic symptoms.

Although this study represents, to the best of our knowledge, one of the largest investigations into the neural correlates of apathy in AD, a few limitations must be mentioned. First, apathy in this sample was assessed using only available NPI-Q data that offer limited information on the duration of symptoms (answers to the NPI-Q refer to the previous month only) and, therefore, prevented the implementation of the diagnostic criteria proposed by Miller and colleagues [6]. A potential strategy to overcome this limitation in future studies using public datasets like ADNI could be to focus on patients presenting with apathy in at least two consecutive assessments (6 to 12 months apart). However, this may lead to a substantial decrease in the sample size and may not ensure that patients presenting with apathy at two consecutive time points had apathy for the whole or most of the time in between assessments. Second, data on biomarkers for AD were missing for a few participants. However, both patient groups were significantly more likely to be positive for both Aβ and p-tau than CU, as expected in samples of patients with MCI/dementia due to AD.

## 5. Conclusions

Apathy in AD appears to be primarily associated with WM damage, mainly in frontal and limbic WM tracts. Patients with AD and apathy tend to have consistently larger volumes of WM hyperintensities and more alterations in WM microstructure, i.e., lower FA and higher AxD and RD. The cause of these WM alterations remains unclear; for instance, the apathetic patients in this study were not more likely to have a history of cardiovascular risk factors than the non-apathetic patients. Further investigations are needed to clarify the complex interplay between biological mechanisms (e.g., AD biomarkers, brain metabolism, structural and functional alterations) implicated in apathy across the AD continuum to support the development of effective intervention strategies [71].

## Figures and Tables

**Figure 1 brainsci-12-01383-f001:**
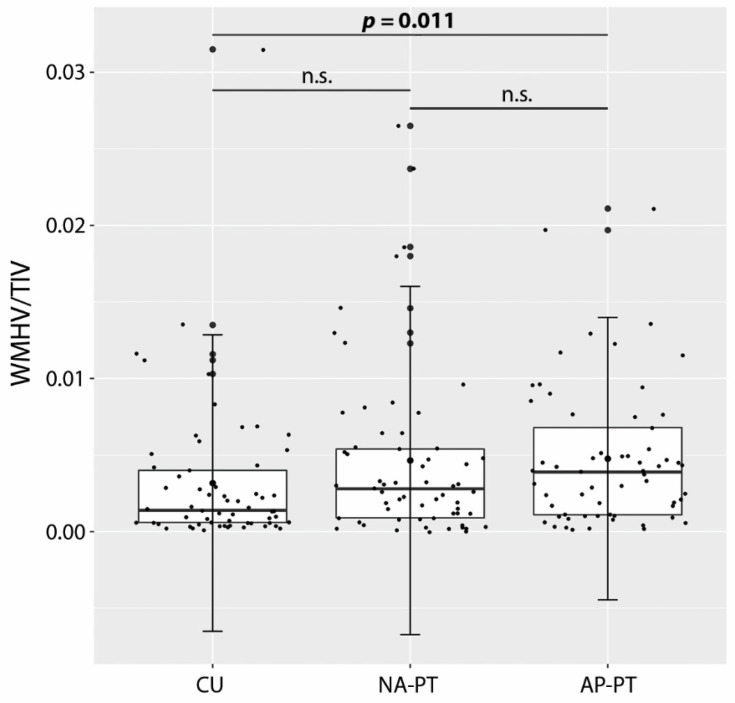
Differences in relative WM hyperintensity volume (WMHV/TIV) across groups (error bars are standard deviations). AP-PT: Patients with apathy, CU: Cognitively unimpaired, NA-PT: Patients without apathy, TIV: Total intracranial volume, WM: White matter, WMHV: White matter hyperintensity volume.

**Figure 2 brainsci-12-01383-f002:**
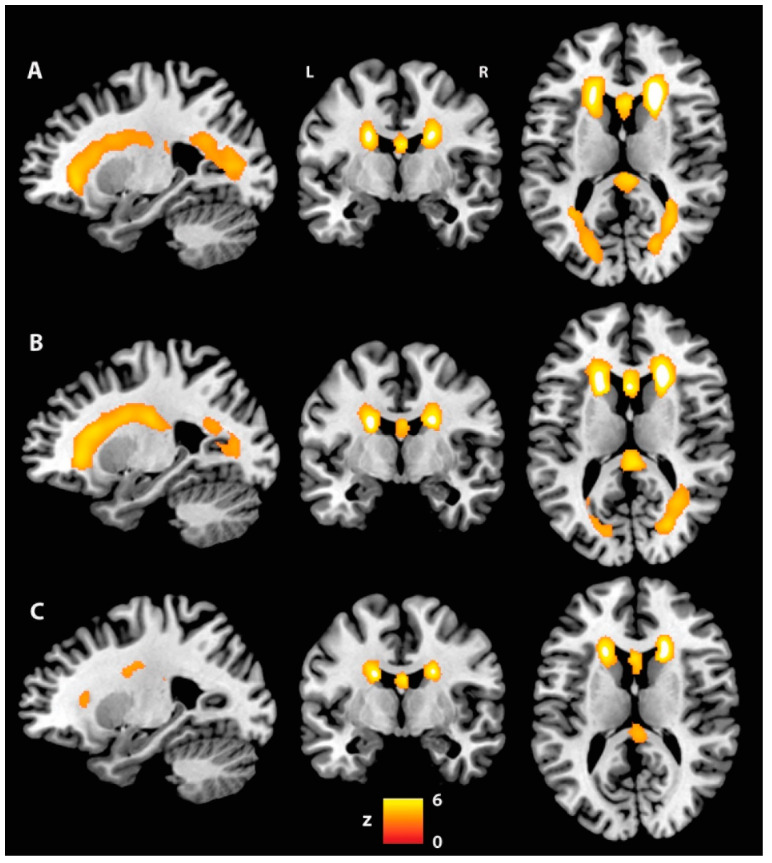
WMH maps created by means of a one-sample *t*-test (cluster-forming threshold *p* < 0.001; FWE-corrected at cluster level): (**A**) AP-PT group; (**B**) NA-PT group; (**C**) CU group. L: Left, R: Right.

**Figure 3 brainsci-12-01383-f003:**
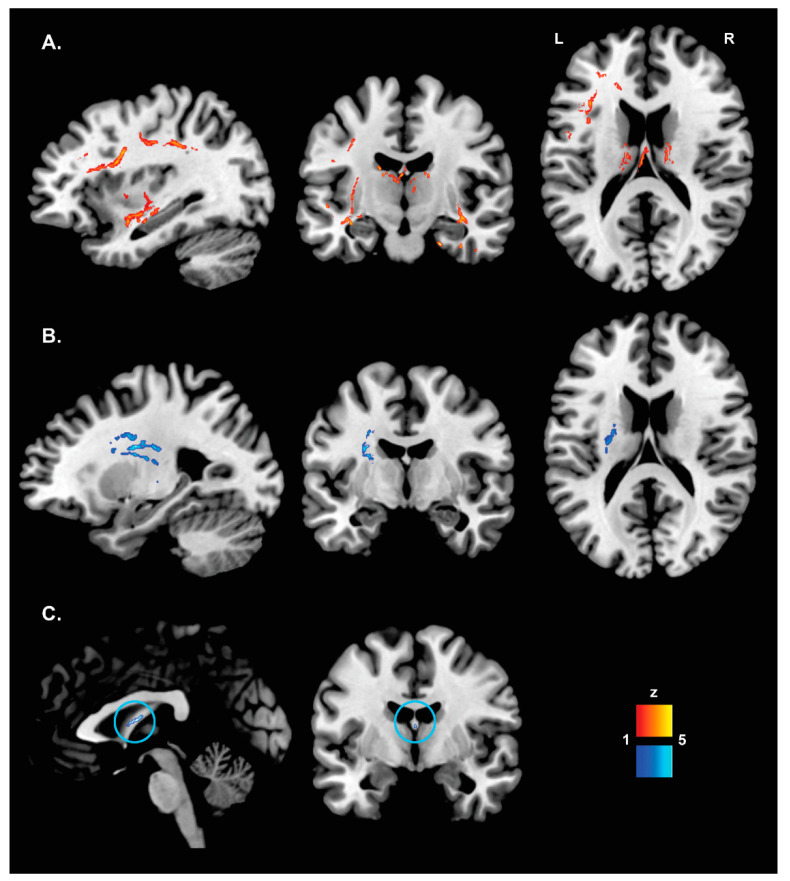
Clusters of altered WM microstructural integrity in the AP-PT compared with the CU group: (**A**) fractional anisotropy—AP-PT < CU; (**B**) axial diffusivity—AP-PT > CU; (**C**) radial diffusivity—AP-PT > CU (the significant small cluster in the fornix is highlighted by the blue circle).

**Table 1 brainsci-12-01383-t001:** Differences in demographic, clinical and neural characteristics across participant groups. Values are means and standard deviations unless otherwise specified.

Characteristics	AP-PT (*n* = 61)	NA-PT (*n* = 61)	CU (*n* = 61)	F	*p*
Age (years)	73.33 (6.97)	73.93 (8.47)	73.11 (6.26)	0.21	0.813
Education (years) ^a^	16.00 (4)	16.00 (5)	16.00 (4)	1.27 ^b^	0.529
Sex (F/M) ^c^	**17/44 ***	**29/32**	33/28	9.27 ^d^	0.010
CDR ^a^	0.00 (0.5) *	0.50 (0) *	0.00 (0)	149.60 ^b^	<0.001
MMSE ^a^	26.00 (4) *	26.00 (3) *	29.00 (2)	50.02 ^b^	<0.001
NPI-Q (total) ^a^	**6.00 (6) ***	**1.00 (2) ***	0.00 (0)	132.16 ^b^	<0.001
NPI-Q total—Apathy score ^a^	**5.00 (6) ***	**1.00 (2) ***	0.00 (0)	106.38 ^b^	<0.001
TIV (ml)	1500.58 (140.57) *	1452.82 (151.20)	1439.58 (123.10)	3.26	0.041
GMV/TIV ^a^	0.39 (0.05) *	0.41 (0.07) *	0.43 (0.05)	20.34 ^b^	<0.001
WMV/TIV ^a^	0.27 (0.03)	0.27 (0.02)	0.27 (0.02)	1.06 ^b^	0.589
WMHV/TIV ^a^	0.004 (0.006) *	0.003 (0.004)	0.001 (0.003)	8.27 ^b^	0.016
Aβ (+/−)	41/17 * (*n* = 58)	32/21 * (*n* = 53)	12/39 (*n* = 51)	26.17 ^d^	<0.001
p-tau (+/−)	38/20 * (*n* = 58)	31/22 * (*n* = 53)	10/41 (*n* = 51)	25.87 ^d^	<0.001

Aβ: Amyloid beta, AP-PT: Patients with apathy, CDR: Clinical Dementia Rating scale, CU: Cognitively unimpaired, F: Females, GMV: Gray matter volume, M: Males, MMSE: Mini Mental State Examination, NA-PT: Patients without apathy, NPI-Q: Neuropsychiatric Inventory Questionnaire, p-tau: phosphorylated tau, TIV: Total intracranial volume, WMHV: White matter hyperintensity volume, WMV: White matter volume. ^a^ Median (interquartile range). ^b^ Kruskal-Wallis test. ^c^ Frequencies. ^d^ Chi-square test. * Patient groups significantly different from the CU group in pairwise *post hoc* comparisons. In bold, significant difference between patient groups in pairwise *post hoc* comparisons.

**Table 2 brainsci-12-01383-t002:** Differences in DTI indices between AP-PT and CU groups (*p*FWE < 0.05).

*p*	Cluster Extent	Side	White Matter Tract	*t*	MNI Coordinates		
					x	y	z
*Fractional anisotropy: AP-PT < CU*							
0.035	3283	-	Fornix	4.65	0	−4	13
		-	Fornix	4.31	0	5	7
		-	Fornix	4.28	0	2	10
		L	Uncinate fasciculus	4.20	−23	−1	−9
		R	ATR	4.20	14	−10	16
		L	ATR	4.20	−13	−5	16
0.040	1546	L	SLF	3.91	−38	21	16
		L	SLF	3.84	−35	2	30
		L	SLF	3.82	−32	5	29
		L	SLF	3.60	−34	6	23
		L	SLF	3.52	−33	−17	38
		L	SLF	3.49	−33	−33	37
0.041	929	R	IFOF	4.34	35	−11	−14
		R	IFOF	4.12	38	−9	−15
		R	ILF	3.42	37	−6	−22
		R	IFOF	3.35	34	−6	−13
		R	Uncinate fasciculus	3.24	30	7	−12
		R	ILF	3.20	41	−7	−36
0.045	177	R	Cingulum (temporal)	5.03	22	−13	−28
		R	Cingulum (temporal)	4.21	29	−29	−17
		R	Cingulum (temporal)	3.96	29	−30	−15
		R	Cingulum (temporal)	3.71	30	−26	−19
		R	Cingulum (temporal)	3.66	25	−24	−20
		R	Cingulum (temporal)	3.66	26	−26	−20
0.050	117	L	Forceps minor	3.03	−19	33	14
		L	Forceps minor	2.93	−15	26	20
		L	Forceps minor	2.92	−17	34	14
		L	Forceps minor	2.53	−16	32	21
		L	Forceps minor	2.52	−17	30	19
		L	Forceps minor	2.48	−16	30	22
0.047	82	R	ILF	4.04	33	−1	−30
		R	ILF	3.56	31	3	−31
		R	Cingulum (temporal)	3.52	34	−5	−31
		R	ILF	3.17	32	0	−28
		R	ILF	2.87	30	0	−29
		R	Cingulum (temporal)	2.78	34	−3	−34
0.049	76	R	Cingulum (temporal)	4.03	36	−17	−28
		R	Cingulum (temporal)	3.42	34	−19	−26
		R	Cingulum (temporal)	3.34	36	−12	−29
		R	Cingulum (temporal)	2.86	36	−20	−25
		R	Cingulum (temporal)	2.77	34	−9	−35
		R	Cingulum (temporal)	2.72	34	−9	−33
0.050	64	L	ATR	3.06	−21	−50	36
		L	ATR	3.04	−22	−47	39
		L	ATR	3.01	−20	−52	36
		L	SLF	2.76	−23	−50	35
0.050	52	L	Forceps minor	3.34	−11	28	−8
		L	Forceps minor	3.06	−12	28	−6
		L	Forceps minor	2.87	−12	30	−5
		L	Forceps minor	2.34	−15	35	−6
0.050	50	L	Forceps minor	3.09	−13	45	−15
		L	Forceps minor	3.02	−14	43	−14
		L	Forceps minor	2.98	−15	42	−11
0.050	45	L	SLF	3.15	−32	27	25
		L	SLF	2.87	−36	29	26
		L	SLF	2.82	−36	31	26
		L	SLF	2.76	−33	27	28
		L	SLF	2.66	−36	27	27
0.050	44	R	Forceps minor	2.79	18	39	−4
		R	Forceps minor	2.65	19	37	−6
		R	Forceps minor	2.62	17	41	−8
		R	Forceps minor	2.57	17	37	−6
0.050	39	L	Forceps minor	3.46	−28	39	17
		L	Forceps minor	3.26	−30	40	14
		L	Forceps minor	2.93	−30	39	18
		L	Forceps minor	2.80	−30	37	17
		L	Forceps minor	2.77	−30	37	15
		L	Forceps minor	2.56	−30	38	12
0.050	32	L	SLF	3.37	−19	−52	54
		L	SLF	3.19	−18	−53	51
		L	SLF	3.00	−18	−49	46
		L	SLF	2.81	−18	−53	48
0.050	27	L	Forceps major	3.92	−26	−56	21
		L	Forceps major	3.54	−23	−56	22
0.050	18	R	Uncinate fasciculus	2.59	20	20	−13
0.050	15	L	SLF	2.36	−31	10	43
		L	SLF	2.19	−31	6	44
		L	SLF	2.16	−31	8	43
		L	SLF	2.13	−31	7	46
*Axial diffusivity: AP-PT > CU*							
0.040	661	L	SLF	4.43	−25	−6	25
		L	SLF	4.03	−25	−6	19
		L	SLF	3.88	−28	−3	23
		L	SLF	3.82	−28	−7	24
		L	SLF	3.78	−24	−15	13
		L	SLF	3.48	−26	−13	16
*Radial diffusivity: AP-PT > CU*							
0.040	16	-	Fornix	4.58	0	−4	13
		-	Fornix	4.14	0	0	12
		-	Fornix	4.05	0	3	9

AP-PT: Patients with apathy, ATR: Anterior thalamic radiations, CU: Cognitively unimpaired, DTI: Diffusion tensor imaging, ILF: Inferior longitudinal fasciculus, IFOF: Inferior fronto-occipital fasciculus, MNI: Montreal Neurological Institute, NA-PT: Patients without apathy, SLF: Superior longitudinal fasciculus.

**Table 3 brainsci-12-01383-t003:** Differences in cognitive performance across participant groups. Values are means and standard deviations unless otherwise specified.

Characteristics	AP-PT (*n* = 61)	NA-PT (*n* = 61)	CU (*n* = 61)	F	*p*
CDT—drawing ^a^	5.00 (1) *	5.00 (1)	5.00 (1)	5.66 ^b^	0.059
CDT—copy ^a^	5.00 (1) *	5.00 (0)	5.00 (0)	9.47 ^b^	0.009
LMT—IR	7.69 (4.41) (*n* = 58) *	7.90 (4.06) *	13.90 (3.05)	50.04	<0.001
LMT—DR ^a^	4.50 (9) * (*n* = 58)	6.00 (8) *	12.00 (5)	71.57 ^b^	<0.001
AVLT—IR	29.00 (10.37) *	30.13 (8.04) *	44.85 (10.03)	52.44	<0.001
AVLT—DR ^a^	0.00 (4) *	2.00 (4) *	8.00 (5)	59.99 ^b^	<0.001
CFT-A (total)	14.31 (5.26) *	15.26 (4.71) *	20.90 (4.90)	31.42	<0.001
CFT-A (perseverations) ^a^	0.00 (1)	0.00 (1)	0.00 (1)	2.49 ^b^	0.288
CFT-A (intrusions) ^a^	0.00 (0)	0.00 (0)	0.00 (0)	0.52 ^b^	0.769
TMT-A (sec) ^a^	41.00 (25) *	40.00 (18) *	34.00 (13)	17.44 ^b^	<0.001
TMT-B (sec) ^a^	116.00 (137) * (*n* = 59)	120 (114) *	76.00 (32)	30.33 ^b^	<0.001
BNT (total) ^a^	27.00 (6) *	26.00 (4) *	29.00 (3)	17.94 ^b^	<0.001

AP-PT: Patients with apathy, AVLT: Auditory Verbal Learning Test, BNT: Boston Naming Test, CDT: Clock Drawing Test, CFT-A: Category Fluency Test—animals, CU: Cognitively unimpaired, DR: Delayed recall, IR: Immediate recall, LMT: Logical Memory Test, NA-PT: Patients without apathy, TMT—A/B: Trail Making Test—part A/part B. ^a^ Median (interquartile range). ^b^ Kruskal-Wallis test. * Patient groups significantly different from the CU group in pairwise *post hoc* comparisons.

## Data Availability

All ADNI data are made publicly available upon request.

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
