# Peer review of "Macrostructural and Microstructural White Matter Alterations Are Associated with Apathy across the Clinical Alzheimer’s Disease Spectrum"

_brainsci, 2022, doi:10.3390/brainsci12101383_

Round 1

Reviewer 1 Report

The authors present an important study showing the association between macrostructural and microstructural white matter (WM) alterations with apathy across the clinical Alzheimer’s disease (AD) spectrum. They correctly state that apathy in AD appears to be primarily associated with WM damage.

l   The approach to data analysis is valid. This is a well written study which will add to our understanding of the implication of apathy across AD.

l   Only one minor recommendation for the authors: please add a separate abbreviation list for the readers to follow in a more comprehensive way.

Author Response

We thank the reviewers for their comments on our manuscript. We have worked on all points raised and amended the manuscript accordingly. All changes, including those to spelling and grammar inconsistencies, have been highlighted in yellow.

Reviewer 1

#1. Only one minor recommendation for the authors: please add a separate abbreviation list for the readers to follow in a more comprehensive way.

Following the reviewer’s suggestion, an abbreviation list has been included in the supplementary material file.

Reviewer 2 Report

This is a reasonable attempt to better understand pathological changes related to apathy in Alzheimer's disease patients. Some clarifications would help the reader better understand the results. The title states macro and microstructural changes; these could be defined in the abstract or the discussion. There are 3 study groups but the text mentions two groups (line 252 and 358). Clarify CU group in Abstract. What screening was done to select 224 subjects out of 770?   What is the small circle in Fig 3C? Line 31 has a comma splice. Line 46, omit first comma. Line 177, were=was. 

Author Response

We thank the reviewers for their comments on our manuscript. We have worked on all points raised and amended the manuscript accordingly. All changes, including those to spelling and grammar inconsistencies, have been highlighted in yellow.

Reviewer 2

#1. The title states macro and microstructural changes, these could be defined in the abstract or the discussion.

We thank the reviewer for this comment. Indices of macrostructural WM damage investigated in this study were WM volume and WM hyperintensity volume. While DTI indices (fractional anisotropy, mean, axial and radial diffusivity) were considered as measured of microstructural WM damage. This has now been specified in the following points throughout the manuscript, i.e., in our Abstract:

AP-PT had increased WM damage, both macrostructurally, i.e., larger WM hyperintensity volume, and microstructurally, i.e., increased radial/axial diffusivity

In the Introduction, where the study aims are stated:

by comparing WM volumes, WMH volumes, as measures of WM macrostructural alterations, and diffusion tensor imaging (DTI) indices, as measures of WM microstructural alterations

And in the Discussion:

(i.e., macrostructural WM alterations) and alterations in DTI indices (i.e., WM microstructural damage)

#2. There are 3 study groups but the text mentions two groups (line 252 and 358). Clarify CU group in Abstract.

The CU group was included in the study as a control group primarily to ensure that both patient groups had neural and cognitive signatures of AD (i.e., medial temporal lobe atrophy and cognitive impairment, especially in long term memory). Moreover, although the main hypothesis was related to potential neural differences in white matter damage between apathetic and non-apathetic patients, comparing each patient group to the control group would enable potential subtle differences difficult to detect by directly comparing two samples of patients matched for most clinical and neural characteristics.

We have specified the role of the CU group as a control group both in the Abstract:

Sixty-one cognitively unimpaired (CU) participants were also included as controls.

The Methods:

2 patient groups, i.e., NA-PT (n = 61) and AP-PT (n = 61) and a control group of CU participants (n = 61)

And the Discussion:

when comparing patient groups with a control group of matched CU older adults

#3. What screening was done to select 224 subjects out of 770?  

The initial sample of 770 ADNI participants was selected based on the availability of T1-weighted, FLAIR and DTI scans. Within this group, we selected those who also had a comprehensive clinical assessment including the following measures (as specified in lines 147-159): 1) NPI-Q score (to define apathetic and non-apathetic patients; NPI-Q score had to be 0 for cognitively unimpaired controls); 2) CDR score (to ascertain clinical diagnosis); 3) MMSE score (to establish the global cognitive status of participants); 4) comprehensive cognitive assessment.

Only 224 participants had a comprehensive clinical assessment. We have now specified this point further in line 160:

After screening, a total of 224 participants met the inclusion criteria (i.e., had a comprehensive clinical assessment as described above)

#4. What is the small circle in Fig 3C?

Since a significant difference in radial diffusivity (RD) between apathetic patients (AP-PT) and cognitively unimpaired (CU) controls, i.e., higher RD values in the AP-PT than in the CU group, was found only in a small cluster of white matter in the fornix, a blue circle was used to highlight this significant result that might have been more difficult to notice in Figure 3. To improve clarity, we have added a specification regarding the purpose of the blue circle in the legend of Figure 3:

(the significant small cluster in the fornix is highlighted by the blue circle)

#5. Line 31 has a comma splice.

The comma in line 31 has been replaced by a full stop.

#6. Line 46, omit first comma.

The comma has now been removed.

#7. Line 177, were=was.

Thank you, this typo has now been amended.